## TRANSLATIONAL PERSPECTIVES

### Probing shear-stress-mediated cerebral vasodilatation in humans – it's a NO brainer

**Jui-Lin Fan** ⓘD **and James P. Fisher** ⓘD

*Manaaki Manawa – The Centre for Heart Research, Department of Physiology, Faculty of Medical & Health Sciences, University of Auckland, New Zealand*

Email: jp.fisher@auckland.ac.nz

Edited by: Kim Barrett & Caroline Rickards

Linked articles: This Translational Perspectives article highlights an article by Hoiland *et al*. To read this paper, visit https://doi.org/10.1113/JP282427.

The peer review history is available in the Supporting Information section of this article (https://doi.org/10.1113/JP282707#support-information-section).

The burden of brain diseases (e.g. stroke, dementia, Alzheimer's disease) is considerable and has grown substantially alongside the increased life expectancy of western industrialised nations. Disrupted cerebrovascular endothelial function contributes significantly to the pathogenesis and neurodegeneration of several of these conditions. The early identification of individuals with or at risk of cerebrovascular impairments, using tests with a robust mechanistic basis, may have clinical utility for guiding interventional strategies, thereby helping to ameliorate disease progression before it becomes firmly established. Some of the earliest measurements of cerebral blood flow in humans quantified the "striking and consistent" steady-state hyperaemic response to the inhalation of carbon dioxide ($CO_2$) (5–7%) (Kety & Schmidt 1948), which is important for pH and thus neuronal homeostasis. Subsequently, the cerebrovascular response to a steady-state $CO_2$ stimulus test has been reported to be diminished in several clinical conditions and has prognostic value (Juttukonda & Donahue 2019). In a recent issue of the *Journal of Physiology*, Hoiland and colleagues provided important new mechanistic insights into a novel test that potentially provides a targeted assessment of cerebrovascular endothelial function.

A transient $CO_2$ stimulus – provided by inspiring a $CO_2$-enriched gas mixture to raise $P_{ETCO_2}$ by +9 mmHg for 30 s – evokes an increase in shear stress within the internal carotid artery (ICA) that peaks at ∼20 s after $CO_2$ onset and is followed by ICA vasodilatation at ∼80 s (Hoiland *et al*. 2017). Since this time course is akin to that observed in other conduit vessels during the flow-mediated dilatation test, the authors discounted any direct actions of $CO_2$ and instead hypothesized that a shear-stress-mediated release of nitric oxide (NO) was responsible for the latent ICA vasodilatation. To explore this possibility, Hoiland *et al*. (2022) re-evaluated the ICA response to the transient $CO_2$ stimulus both with a saline control and i.v. infusion of the non-selective NO synthase inhibitor, $N^G$-monomethyl-L-arginine (L-NMMA, 5 mg/kg bolus and 50 $\mu$g/kg/min maintenance dose). Notably, L-NMMA abolished the increase in the trans-cerebral release of nitrite (index of NO production) during the transient $CO_2$ stimulus and reduced the increase in ICA vasodilatation (by ∼37%), while shear stress was not different between conditions. Such observations were interpreted as revealing the integral role of shear-stress-mediated endothelial NO release to the cerebrovascular response to a transient $CO_2$ stimulus. In contrast, an NO synthase mechanism was not observed to be obligatory for the cerebrovascular response to steady-state $CO_2$ stimulus. Indeed, the magnitude of the ICA response, along with that of the vertebral artery (VA) and middle cerebral artery blood velocity (MCAv), to steady-state hypercapnia (5 min at +4.5 and +9 mmHg $P_{aCO_2}$) was not different in the L-NMMA and saline conditions.

The observed failure of L-NMMA to diminish steady-state cerebrovascular $CO_2$ reactivity is perhaps surprising given the available *in vitro* and *in vivo* evidence supporting the role of endothelial-derived NO in $CO_2/H^+$-mediated cerebral vasodilatation (Yoon *et al*. 2012). Indeed, in patients with cardiovascular risk factors, diminished steady-state cerebrovascular $CO_2$ reactivity ($\Delta MCAv/\Delta P_{ETCO_2}$) is reportedly associated with impaired peripheral vascular NO signalling (Lavi *et al*. 2006), while L-arginine infusion (a precursor of NO) restores the blunted steady-state cerebrovascular $CO_2$ reactivity (Zimmermann & Haberl 2003). These discrepancies may relate to the differing primary outcome variables (ICA diameter *vs*. MCAv) and study cohorts, along with the redundancy in mechanisms controlling cerebral blood flow. Nevertheless, for investigators evaluating strategies by which the NO synthase-dependent pathway could be targeted to enhance cerebrovascular endothelial function (e.g. nutrition, physical activity), the transient $CO_2$ test may provide superior specificity over traditional steady-state methods.

Practical and scientific advantages of the transient $CO_2$ over the steady-state approach include the shorter exposure perhaps more readily lending itself to use in higher-risk patients and limiting confounding side effects (e.g. hypercapnia-induced sympathetic nervous system activation). While the technical set-up required to assess the cerebrovascular response to a transient $CO_2$ stimulus is impressively sophisticated, it could pose a significant barrier to the wider adoption of this approach in both research and clinical settings. Irrespectively, the application of this technique in relevant patient groups (e.g. those at risk of, or with established, cerebrovascular complications) and the exploration of its prognostic utility is important, as is determining how its sensitivity/specificity compares to steady-state cerebrovascular $CO_2$ reactivity. No doubt there will also be interest in describing how changing physiological conditions (e.g. heat, exercise, mental stress, hypoxia), both acutely and chronically, affect the cerebrovascular response to a transient $CO_2$ stimulus in a variety of groups.

In summary, Hoiland *et al*. (2022) utilize a cutting-edge experimental approach to provide valuable mechanistic insights into a novel modality for assessing cerebrovascular endothelial function. The demonstration that the shear-stress-mediated vasodilatation in an extracranial vessel (i.e. ICA) provoked by a transient $CO_2$ stimulus is substantively dependent on NO synthase activity is valuable. Basic studies to elucidate other contributing vasoactive mediators (e.g. prostaglandins) and applied studies in relevant patient populations to quantify cerebrovascular endothelial function and assess the prognostic utility of this novel marker would be exciting future directions for exploration.

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

## Additional information

### Competing interests

The authors declare that they have no competing financial interests.

### Author contributions

J.-L. F.: Conception or design of the work; Drafting the work or revising it critically for important intellectual content. J. F.: Conception or design of the work; Drafting the work or revising it critically for important intellectual content; Both authors have read and approved the final version of this manuscript and agree to be accountable for all aspects of the work in ensuring that questions related to the accuracy or integrity of any part of the work are appropriately investigated and resolved. All persons designated as authors qualify for authorship, and all those who qualify for authorship are listed.

### Funding

J.P.F. is supported by the Health Research Council of New Zealand, the Marsden Fund Royal Society Te Apārangi and the Auckland Medical Research Foundation.

### Acknowledgement

The authors are grateful to Dr Bengt Kayser for his thoughtful comments during the development of the manuscript.

Open access publishing facilitated by The University of Auckland, as part of the Wiley – The University of Auckland agreement via the Council of Australian University Librarians.

### Keywords

blood flow, brain, human, nitric oxide

### Supporting information

Additional supporting information can be found online in the Supporting Information section at the end of the HTML view of the article. Supporting information files available:

**Peer Review History**

