## [Peer Review History · The Journal of Physiology]

Probing shear-stress mediated cerebral vasodilatation in humans - it's a NO brainer

Jui-Lin Fan and James P Fisher
DOI: 10.1113/JP282707

Corresponding author(s): James Fisher (jp.fisher@auckland.ac.nz)

The following individual(s) involved in review of this submission have agreed to reveal their identity: Ryan L Hoiland (Referee #1)

Review Timeline:

Submission Date:	16-Dec-2021
Editorial Decision:	22-Dec-2021
Revision Received:	22-Dec-2021
Accepted:	04-Jan-2022

Senior Editor: Kim Barrett

Reviewing Editor: Caroline Rickards

Transaction Report:

Dear Dr Fisher,

Re: JP-TP-2021-282707 "Probing shear-stress mediated cerebral vasodilation in humans - it's a NO brainer" by Jui-Lin Fan and James P Fisher

Thank you for submitting your manuscript to The Journal of Physiology. It has been assessed by a Reviewing Editor and by 1 expert referees and I am pleased to tell you that it is considered to be acceptable for publication following satisfactory revision.

The reports are copied at the end of this email. Please address all of the points and incorporate all requested revisions, or explain in your Response to Referees why a change has not been made.

NEW POLICY: In order to improve the transparency of its peer review process The Journal of Physiology publishes online as supporting information the peer review history of all articles accepted for publication. Readers will have access to decision letters, including all Editors' comments and referee reports, for each version of the manuscript and any author responses to peer review comments. Referees can decide whether or not they wish to be named on the peer review history document.

I hope you will find the comments helpful and have no difficulty returning revisions within 4 weeks.

If you need to check to make sure that your Methods section conforms to the principles of UK regulations, you may wish to refer to Grundy (2015):

Grundy (2015) J. Physiol. 2015 Jun 15;593(12):2547-9 <https://doi.org/10.1113/JP270818>

Your revised manuscript should be submitted online using the links in Author Tasks Link Not Available. This link is to the Corresponding Author's own account, if this will cause any problems when submitting the revised version please contact us.

The image files from the previous version are retained on the system. Please ensure you replace or remove any files that have been revised.

REVISION CHECKLIST:

- Summary data must be reported as mean {plus minus} SD or 95% confidence interval
- All table and figure legends with summary data must include the statistical test used in the table/figure and sample size
- Figures with summary data bars must include individual data points, or box whisker plots when $n > 30$.
- Article file, including any tables and figure legends, must be in an editable format (eg Word)
- Upload each figure as a separate high quality file
- Upload a full Response to Referees, including a response to any Senior and Reviewing Editor Comments;
- Upload a copy of the manuscript with the changes highlighted.

- A potential 'Cover Art' file for consideration as the Issue's cover image;
- Appropriate Supporting Information (Video, audio or data set https://jp.msubmit.net/cgi-bin/main.plex?form_type=display_requirements#supp).

To create your 'Response to Referees' copy all the reports, including any comments from the Senior and Reviewing Editors, into a Word, or similar, file and respond to each point in colour or CAPITALS and upload this when you submit your revision.

I look forward to receiving your revised submission.

If you have any queries please reply to this email and the Peer Review Coordinator will be pleased to advise.

If revision is not possible, or if you cannot respond to the requests for change, contact us by return email as soon as

possible, giving reasons for the difficulties. Withdrawal of the manuscript may be necessary in these circumstances, and instruction will be given on how to proceed. Please note that a paper must be withdrawn before it can be submitted to another journal. If any issues remain unresolved please contact the Publications Office at jphysiol@physoc.org

If you would like help with English language editing, or other article preparation support, Wiley Editing Services offers expert help with English Language Editing, as well as translation, manuscript formatting, and figure formatting at www.wileyauthors.com/eeo/preparation. You can also check out our resources for Preparing Your Article for general guidance about writing and preparing your manuscript at www.wileyauthors.com/eeo/prepresources.

Yours sincerely,

Professor Kim E. Barrett
Editor-in-Chief
The Journal of Physiology
<https://jp.msubmit.net>
<http://jp.physoc.org>
The Physiological Society
Hodgkin Huxley House
30 Farringdon Lane
London, EC1R 3AW
UK
<http://www.physoc.org>
<http://journals.physoc.org>

EDITOR COMMENTS

Reviewing Editor:

Thank you for your perspectives article. Please review and respond to the comments provided by Dr. Hoiland.

REFEREE COMMENTS

Referee #1:

Dear Drs. Fan & Fisher

Thank you for taking the time to write this perspective on our recently published article in the Journal of Physiology. I very much enjoyed reading this perspective and feel it appropriately reflects the observations made in our experiments. The corrections I suggest below for Lines 38, 40, 47, and 84 are important for the accuracy of the perspective, while my comment on the paragraph citing the studies by Zimmermann and Lavi is simply some thoughts that you can take or leave at your discretion. Thank you again for taking the time to provide your perspective on our paper.

Sincerely,

Ryan Hoiland

Comments...

Line 38: The transient CO₂ stimulus is targeted to +9mmHg above baseline for 30-seconds (rather than the 13 noted in text). A greater CO₂ stimulus doesn't massively alter the shear stimulus, but greatly influences comfort of the research volunteer (determined through pilot testing).

Line 40: While the Carter 2016 study was instrumental in the progression of our work on shear-mediated dilation of the internal carotid artery, it did not include the transient CO₂ test, but rather looked at time courses with a steady-state stimulus, providing preliminary evidence for shear-mediated vasodilation due to the observed time course being analogous to that of peripheral FMD. It was in the Hoiland et al., 2017, AJP Heart Circ Physiol paper that we reported the time delays for shear and vasodilation with the transient test. This distinction is important and requires the correct citation, as other investigators have since misinterpreted the Carter et al., paper as it relates to the transient test. For example, some laboratories have interpreted the maximum internal carotid artery dilation with steady state CO₂ as shear-mediated vasodilation (based on the Carter paper), however, as outlined in figure 6 of our paper this is not the case. We hope that this new paper will help resolve this. Citation: Hoiland et al., (2017). Am J Physiol Heart Circ Physiol. 313(1):H24-H31.

Line 47: trans-cerebral "nitrite", not nitrAte. Importantly, exchange of nitrAte across a vascular bed has not been shown as indicative of endothelial NO release, whereas exchange of nitrite has. Citation: Lauer et al., (2001). Proc Natl Acad Sci USA. 98(22):12814-9.

Line 84: we have only quantified shear mediated vasodilation in the ICA (we did not assess this in the VA).

In the paragraph discussing how NO has demonstrated a role in other studies (e.g. Zimmermann study), an important distinction is that the transient CO₂ test is measuring a diameter response, not a flow response. This is a macrovascular outcome rather than what I would assume to be a primarily microvascular outcome (flow). Indeed, the reactive hyperemia did not differ with L-NMMA vs Saline, indicating that these micro vs. macrovascular responses provide us with different information. The distinction between "flow" and "diameter" outcomes may therefore warrant mention here. This may be even more relevant given the study cited by Lavi measured post-occlusion forearm blood flow AUC rather than peak dilation of the brachial artery (i.e. FMD), which also seemingly reflects differing physiology. While I know I am being rather pedantic (apologies!), I do think these nuanced distinctions are important to make.

END OF COMMENTS

Confidential Review

16-Dec-2021

Referee #1:

“Dear Drs. Fan & Fisher

Thank you for taking the time to write this perspective on our recently published article in the Journal of Physiology. I very much enjoyed reading this perspective and feel it appropriately reflects the observations made in our experiments. The corrections I suggest below for Lines 38, 40, 47, and 84 are important for the accuracy of the perspective, while my comment on the paragraph citing the studies by Zimmermann and Lavi is simply some thoughts that you can take or leave at your discretion. Thank you again for taking the time to provide your perspective on our paper.

Sincerely,

Ryan Hoiland”

Response: Thank you for the kind words and support for our perspective. The corrections have all been applied as suggested. We apologise for our oversights. Congratulations again for an excellent study.

“Comments...

Line 38: The transient CO₂ stimulus is targeted to +9mmHg above baseline for 30-seconds (rather than the 13 noted in text). A greater CO₂ stimulus doesn't massively alter the shear stimulus, but greatly influences comfort of the research volunteer (determined through pilot testing).”

Response: Typo corrected.

“Line 40: While the Carter 2016 study was instrumental in the progression of our work on shear-mediated dilation of the internal carotid artery, it did not include the transient CO₂ test, but rather looked at time courses with a steady-state stimulus, providing preliminary evidence for shear-mediated vasodilation due to the observed time course being analogous to that of peripheral FMD. It was in the Hoiland et al., 2017, AJP Heart Circ Physiol paper that we reported the time delays for shear and vasodilation with the transient test. This distinction is important and requires the correct citation, as other investigators have since misinterpreted the Carter et al., paper as it relates to the transient test. For example, some laboratories have interpreted the maximum internal carotid artery dilation with steady state CO₂ as shear-mediated vasodilation (based on the Carter paper), however, as outlined in figure 6 of our paper this is not the case. We hope that this new paper will help resolve this. Citation: Hoiland et al., (2017). Am J Physiol Heart Circ Physiol. 313(1):H24-H31.”

Response: Carter et al., replaced by Hoiland et al.

“Line 47: trans-cerebral "nitrite", not nitrAte. Importantly, exchange of nitrAte across a vascular bed has not been shown as indicative of endothelial NO release, whereas exchange of nitrite has. Citation: Lauer et al., (2001). Proc Natl Acad Sci USA. 98(22):12814-9.”

Response: Typo corrected.

Line 84: we have only quantified shear mediated vasodilation in the ICA (we did not assess

this in the VA).

Response: Typo corrected.

“In the paragraph discussing how NO has demonstrated a role in other studies (e.g. Zimmermann study), an important distinction is that the transient CO₂ test is measuring a diameter response, not a flow response. This is a macrovascular outcome rather than what I would assume to be a primarily microvascular outcome (flow). Indeed, the reactive hyperemia did not differ with L-NMMA vs Saline, indicating that these micro vs. macrovascular responses provide us with different information. The distinction between “flow” and “diameter” outcomes may therefore warrant mention here. This may be even more relevant given the study cited by Lavi measured post-occlusion forearm blood flow AUC rather than peak dilation of the brachial artery (i.e. FMD), which also seemingly reflects differing physiology. While I know I am being rather pedantic (apologies!), I do think these nuanced distinctions are important to make.”

Response: I agree that a difference in the main outcome variable is an important consideration and now mention this, as suggested.

It is intriguing that Zimmerman et al., observed the diminished MCAv responses to ~3 min 5% CO₂/95%O₂ mixture in patients with cardiovascular risk factors to be corrected by L-Arginine. I think it is fair to say that further work is required to better understand if impairments in NO signalling do contribute to cerebro-(micro)vascular dysfunction in clinical populations and if this remains a valuable therapeutic target under certain circumstances.

JPF

Dear Dr Fisher,

Re: JP-TP-2021-282707R1 "Probing shear-stress mediated cerebral vasodilatation in humans - it's a NO brainer" by Jui-Lin Fan
James P Fisher

I am pleased to tell you that your invited Translational Perspectives has been accepted for publication in The Journal of Physiology, subject to any modifications that may be required by the Journal Office to conform to House rules.

NEW POLICY: In order to improve the transparency of its peer review process The Journal of Physiology publishes online as supporting information the peer review history of all articles accepted for publication. Readers will have access to decision letters, including all Editors' comments and referee reports, for each version of the manuscript and any author responses to peer review comments. Referees can decide whether or not they wish to be named on the peer review history document.

The last Word version of the paper submitted will be used by the Production Editors to prepare your proof. When this is ready you will receive an email containing a link to Wiley's Online Proofing System. The proof should be checked and corrected as quickly as possible.

All queries at proof stage should be sent to tjp@wiley.com

Thank you for your contribution to The Journal of Physiology.

Yours sincerely,

Professor Kim E. Barrett
Editor-in-Chief
The Journal of Physiology
<https://jp.msubmit.net>
<http://jp.physoc.org>
The Physiological Society
Hodgkin Huxley House
30 Farringdon Lane
London, EC1R 3AW
UK
<http://www.physoc.org>
<http://journals.physoc.org>

EDITOR COMMENTS:

Caroline A Rickards (Reviewing Editor):

Thank you for addressing the reviewer comments so promptly. Congratulations on this perspectives article.

REFEREE COMMENTS:

Referee #1:

Thank you for your perspective article and the associated revisions. All comments have been addressed and I believe this piece will stimulate important discussions within the field.

END OF COMMENTS

1st Confidential Review

22-Dec-2021